# The impact of physical activity on subjective well-being: The mediating role of exercise identity and the moderating role of health consciousness

Hong-cheng Cui, Yu Zhou [ID]*

College of Leisure and Digital Sports, Guangzhou Sport University, Guangzhou, China

* 1045256659@qq.com

## Abstract

This study investigates the relationship between physical exercise and subjective well-being through a moderated mediation model. It examines how physical exercise influences subjective well-being, focusing on the mediating role of exercise identity and the moderating role of health awareness. The results show that physical exercise significantly enhances subjective well-being, with exercise identity fully mediating this relationship. Additionally, health awareness positively moderates both the relationship between physical exercise and exercise identity and the mediating effect of exercise identity on the link between physical exercise and subjective well-being. This research reveals the mechanisms through which physical exercise promotes well-being and provides theoretical support for advancing national fitness initiatives and improving public health.

## Introduction

The ancient Greek philosopher Aristotle once stated that happiness is the ultimate purpose of life. The pursuit of happiness is a timeless theme for humanity, with all human efforts directed toward achieving it. Subjective well-being refers to a psychological state characterized by individuals' overall evaluation of their quality of life, which results in positive emotions. It is somewhat subjective, stable, and holistic [1]. Collins et al. [2]. found that sustained participation in exercise allows individuals to experience more excitement and pleasure, relieving stress and reducing negative emotions, thereby enhancing their sense of well-being. Physical activity not only alleviates negative emotions but also improves physical fitness, contributing to increased life satisfaction [3, 4]. Regular physical activity leads to a higher quality of life and greater life satisfaction, resulting in more happiness and well-being for those who engage in regular physical activity than for those who rarely participate in such activity [5]. In summary, extensive research has demonstrated the positive impact of physical activity on promoting physical and mental health, life satisfaction, quality of life, and overall happiness.

**Data availability statement:** All relevant data are within the paper and its Supporting Information files.

**Funding:** This work was funded through the following grants: General project of Social Science Planning Project of Guangdong Province in 2024 (GD24CTY11) (Awarded to H.C.C); 2023 Guangdong Provincial Education Science Planning Project (Higher Education Special) (2023GXJK353) (Awarded to H.C.C); 2024 Guangdong Province Graduate Education Innovation Program (2024ANLK_062) (Awarded to H.C.C).

**Competing interests:** The authors have declared that no competing interests exist.

### The role of exercise identity in mediating the relationship between physical activity and well-being

In addition to physical activity, exercise identity is a significant predictor of well-being [6, 7]. Exercise identity refers to the extent to which a person defines his or her self-concept as an exerciser or someone who engages in regular physical activity. Sheldon & Bettencourt [8] noted that the group identity that individuals form in their "everyday world" can effectively enhance their subjective well-being. Exercise identity reflects, to some extent, an individual's attitude toward the role and behavior of physical exercise. Those with a strong exercise identity tend to have scientific and reasonable perceptions of the value of physical exercise. They recognize the benefits of physical activity for physical and mental health and for personal development. People with a strong exercise identity generally view the value and efficacy of physical activity positively and rationally. They also exhibit a sense of belonging, pride in their behavior, and strong self-identity. Consequently, they strive for self-improvement through repeated physical activity, which leads to increased self-esteem, confidence, satisfaction, and happiness [9].

Research has shown a positive correlation between exercise identity and physical activity. Individuals with strong exercise identities tend to be more positive, active, and optimistic than those with weaker exercise identities [10,11]. Physical activity is an embodied practice, and "embodied cognition" emphasizes experiential, emotional, and cognitive interactions with the body [12]. People who are active and regularly engage in physical activity typically develop a stronger sense of group affiliation and exercise identity either with the activity itself or with the group in which they participate [7]. Sustained participation in physical activity fosters the development of exercise identity, which in turn strengthens an individual's self-perception, enhances self-confidence, and improves feelings of achievement, success, social interaction, and belonging. It also adds novelty and enjoyment to life [13]. On this basis, the current study hypothesizes that exercise identity mediates the effect of physical activity on subjective well-being (Hypothesis 1).

### The moderating role of health consciousness in the relationship between physical activity and well-being

Health consciousness refers to the degree of attention that an individual pays to health issues in daily activities, manifested as the awareness and practice of a healthy lifestyle. Individuals with high levels of health consciousness are more likely to engage in healthy behaviors (such as physical exercise and a balanced diet) and are more inclined to recognize the value of physical activity for physical and mental well-being, thereby fostering a stronger exercise identity [14,15].

According to existing research, individuals with high health consciousness place greater emphasis on physical exercise, can better experience its benefits, and thus strengthen their exercise identity. In contrast, individuals with low health consciousness may lack sufficient motivation and awareness, resulting in a weaker exercise identity [16]. Furthermore, we propose that health consciousness not only affects the direct relationship between physical activity and exercise identity but also indirectly

influences subjective well-being by moderating the mediating effect of exercise identity. The theoretical model proposed in this study (as shown in Fig 1) is a moderated mediation model. In this model, the level of health consciousness moderates two pathways: the effect of physical activity on exercise identity and the indirect influence of exercise identity on subjective well-being.

Accordingly, the following hypotheses are proposed:

Hypothesis 2: Health consciousness moderates the relationship between physical activity and exercise identity.

Hypothesis 3: Health consciousness moderates the mediating role of exercise identity in the relationship between physical activity and subjective well-being.

## Method

### Participants

In this study, convenience and purposive sampling methods were employed, and 630 adults from seven cities in China (Changchun, Harbin, Lanzhou, Guiyang, Nanchang, Changsha, and Guangzhou) were recruited. From 05/01/2024–20/03/2024, a total of 630 questionnaires were distributed, and 609 were returned, yielding a response rate of 96.7%. After 25 invalid responses were excluded, 584 valid questionnaires remained, resulting in an effective response rate of 92.7%.

Among the respondents, 58.6% were male, and 41.4% were female. The average age of the participants was M = 35.4 years, SD = 9.6, ranging from 18 to 55 years. With respect to the respondents' educational background, 6.6% had a high school education or below, 12.7% had a specialized education, 62.9% held a bachelor's degree, 15.6% had a master's degree, and 2.2% held a doctoral degree. In terms of marital status, 62.5% of the respondents were married, 34.8% were single, and 2.7% were divorced or widowed. The average annual household income was M = 112,300 CNY, SD = 45,200, with 19.5% earning less than 50,000 CNY, 36.8% earning between 50,000 and 100,000 CNY, 28.6% earning between 100,000 and 200,000 CNY, and 15.1% earning above 200,000 CNY. All procedures performed in this study were approved by the Guangzhou Sports University Ethics Committee. Written informed consent was obtained from all participants prior to their involvement, and the study's purpose, procedures, potential risks, and confidentiality measures were explained to the participants. Furthermore, the participants were informed that their participation was voluntary, that they could withdraw at any time without any penalty, and that all data collected would be kept confidential and used solely for research purposes. All participants in this study were 18 years of age or older; thus, parental or guardian consent was not required.

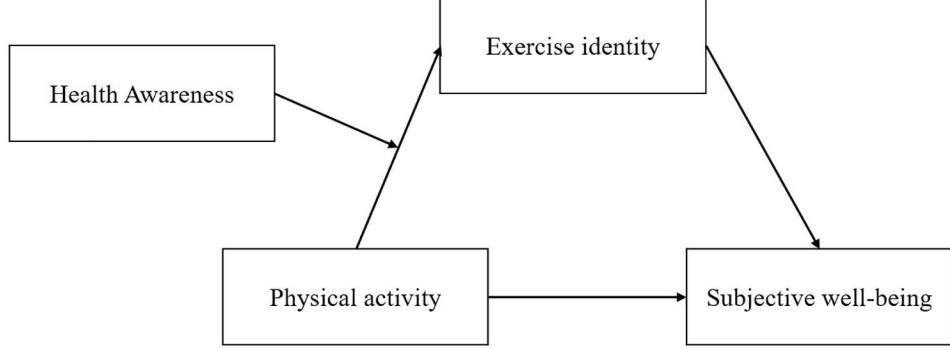

**Fig 1. Theoretical model.**

## Measures

**Physical activity rating scale.** This study used the Physical Activity Rating Scale [17] to measure the participants' physical activity. This scale assesses physical activity behavior by evaluating the frequency, intensity, and duration of physical activity per week. The formula for calculating the level of physical activity is weekly exercise frequency × exercise intensity × duration of each session, with a score ranging from 0 to 100. Each of the three indicators is divided into five levels. Both weekly exercise frequency and intensity are categorized into five grades, with scores ranging from 1 to 5, while the duration of each session is categorized into five levels, with scores ranging from 0 to 4. The Cronbach's alpha for this scale is 0.751.

**Exercise identity scale.** The Exercise Identity Scale, developed by Deci and Ryan [18], was also employed. This scale consists of nine items, such as "I consider myself a person who enjoys physical exercise." The Cronbach's alpha for this scale is 0.940.

**Subjective happiness scale.** Subjective well-being was measured using the Subjective Happiness Scale [19], which contains four items, such as "Overall, I consider myself a (very unhappy - very happy) person." The Cronbach's alpha for this scale is 0.727.

**Health awareness scale.** Health awareness was assessed using the Health Awareness Scale developed by Dutta-Bergman [20]. This scale includes five items, such as "It is very important to me to live as healthy as I can." The Cronbach's alpha for this scale is 0.906.

In this study, all scales except for the Physical Activity Rating Scale were scored on a 7-point Likert scale.

## Reliability and validity testing

Reliability and validity tests are essential prerequisites for ensuring the accuracy of research findings. For the reliability test, Cronbach's alpha was used to assess the internal consistency of the scales. The results of this study show that the Cronbach's alpha coefficients for all scales exceed 0.7, indicating a high level of reliability.

For the validity test, both content validity and discriminant validity were evaluated. In terms of content validity, all the scales used in this study are well established, having been developed by previous researchers and widely applied in related studies, demonstrating strong content validity. For discriminant validity, AMOS 24.0 software was employed to conduct confirmatory factor analysis (CFA) on the four variables: physical activity, health awareness, exercise identity, and subjective well-being. The fit indices of the four-factor model were compared (see Table 1). As shown in Table 1, the four-factor model demonstrated the best fit, with the following indices: $\chi^2$ = 797.848, DF = 181, $\chi^2$/df = 4.408, GFI = 0.928, TLI = 0.917, IFI = 0.928, GFI = 0.876, and RMSEA = 0.076. These results indicate that the four variables have good discriminant validity and represent distinct constructs.

**Table 1. Confirmatory factor analysis.**

| Model | Contained factors | χ2/df | GFI | IFI | TLI | CFI | RMSEA |
|---|---|---|---|---|---|---|---|
| Model 1 | SE, SW, EI, HC | 10.110 | 0.703 | 0.804 | 0.777 | 0.804 | 0.125 |
| Model 2 | SE, (SW+EI+HC) | 9.294 | 0.717 | 0.823 | 0.797 | 0.822 | 0.119 |
| Model 3 | (SE+SW), (EI+HC) | 7.344 | 0.767 | 0.865 | 0.845 | 0.865 | 0.104 |
| Model 4 | SE+SW+EI+HC | 4.408 | 0.876 | 0.928 | 0.917 | 0.928 | 0.076 |

Note: SE means physical activity, SW means subjective well-being, EI means exercise identity, and HC means health consciousness. Model 1 assumes that physical activity, health consciousness, exercise identity, and subjective well-being are four mutually independent latent variables, thus comprising four factors. Model 4 assumes that these four variables can be grouped into a single latent variable, represented as a single-factor model.

# Results

## Common method bias

The questionnaire in this study included trap questions and reverse-scored items to help minimize the risk of common method bias. Additionally, Harman's single-factor test was conducted, and the results show that the total variance explained by the first unrotated principal component is 25.82%, which is below the critical threshold of 40%. This result indicates that common method bias is not a significant issue in this study.

## Correlation analysis

The results of the partial correlation analysis are presented in Table 2. Significant correlations between physical activity, health awareness, exercise identity, and subjective well-being are found, providing initial evidence in support of the theoretical hypotheses (see Table 2). In Table 2, the correlation between physical activity, health awareness and health identity is close to 0.6, and the variables are almost interchangeable. Although we tested the discriminant validity of the variables using confirmatory factor analysis (CFA) (see Table 1), the results show that the four-factor model has significantly better fit indices than the other models do, indicating that physical activity, health awareness, and exercise identity have good discriminant validity statistically. To further examine potential multicollinearity issues among the variables, we calculate the variance inflation factor (VIF). The results indicate that all the independent variables have VIF values less than 5 (physical activity = 1.497, health awareness = 1.561, exercise identity = 2.043), suggesting that multicollinearity is not a significant concern. Therefore, the conclusions of our analysis are not affected by the relatively high correlations among the variables.

## Hypothesis testing

This study uses hierarchical regression and bootstrapping (5000) for theoretical hypothesis testing. The results of the hypothesis testing are shown in Table 3–4 and Fig 2 below.

The relevant theoretical hypotheses were tested following standard mediation effect procedures. First, the effect of the independent variable (physical exercise) on the dependent variable (subjective well-being) was examined. As shown in Model 2 (Table 3), physical exercise has a positive and significant effect on subjective well-being ($\beta = 0.203$, $p < 0.001$). Second, the effect of the independent variable (physical activity) on the mediating variable (exercise identity) was tested. The results from Model 4 indicate that physical activity positively and significantly affects exercise identity ($\beta = 0.552$, $p < 0.001$). Third, the effect of the mediating variable (exercise identity) on the dependent variable (subjective well-being) was examined. The data from Model 3 reveal that exercise identity positively and significantly influences subjective well-being ($\beta = 0.309$, $p < 0.001$). Finally, the joint effect of the independent variable (physical activity) and the mediating variable (exercise identity) on subjective well-being was assessed. In Model 5, after accounting for the mediating effect of exercise identity, the effect of physical exercise on subjective well-being becomes nonsignificant ($\beta = 0.046$, $p > 0.05$),

**Table 2. Correlation coefficients between variables.**

| Variables | 1 | 2 | 3 | 4 |
|---|---|---|---|---|
| 1 Physical activity | 1 | | | |
| 2 Health awareness | .351** | 1 | | |
| 3 Exercise identity | .546** | .603** | 1 | |
| 4 Subjective well-being | .195** | .385** | 300** | 1 |

*Note*:

**indicates $p < 0.01$ (two-tailed test)

**Table 3. Results of hypothesis testing.**

| Variable types | Variables | Exercising Identity Subjective Well-Being | | | | | |
|---|---|---|---|---|---|---|---|
| | | Model 1 | Model 2 | Model 3 | Model 4 | Model 5 | Model 6 |
| Control variables | Sex | -.122* | -.083* | -0.048 | -0.134*** | -0.45 | -0.140*** |
| | Age | 0.020 | 0.027 | 0.012 | 0.043 | 0.014 | -0.003 |
| | Educational level | 0.000 | 0.045 | 0.040 | -0.006 | 0.047 | 0.003 |
| Independent variables | Physical activity | | 0.203*** | 0.309*** | 0.552*** | 0.046 | 0.372*** |
| Mediating variable | Exercise identity | | | | | 0.285*** | |
| Moderating variable | Health awareness | | | | | | 0.458*** |
| Interaction term | Physical activity* Health awareness | | | | | | 0.060* |
| | ΔR2 | 0.010 | 0.046 | 0.098 | 0.347 | 0.097 | 0.528 |
| | F | 3.008** | 8.034*** | 16.760*** | 78.334*** | 13.577*** | 109.584*** |

Note:

*represents $p<0.05$,

**represents $p < 0.01$,

***represents $p<0.001$

**Table 4. Moderated mediation effect test results.**

| Effect | Intermediary path | Health awareness | Ratio | 95% confidence interval | |
|---|---|---|---|---|---|
| | | | | lower limit | upper limit |
| Indirect effect | Exercise identity | | | | |
| | | High | 0.4292 | 0.3541 | 0.5044 |
| | | Low | 0.3141 | 0.2210 | 0.4073 |
| | | Variance | 0.3717 | 0.3074 | 0.4360 |

whereas the effect of exercise identity remains significant (β = 0.285, $p < 0.001$). This finding indicates that exercise identity fully mediates the relationship between physical activity and subjective well-being.

To further validate the mediating effect, the study employed the bootstrap method with 5000 resamples using the Process plug-in in SPSS. The results indicate that the indirect effect of exercise identity on the relationship between physical exercise and subjective well-being is 0.157, with a 95% confidence interval of [0.0947, 0.2205], which does not contain 0, confirming the presence of a significant mediating effect.

To further test Hypotheses 2 and 3, following the recommendation of Edwards et al. (2007), this study utilized the Process plug-in in SPSS with the bootstrap method (5000 resamples) for analysis. As shown in Table 4, a difference in the indirect effect of exercise identity is observed at high and low levels of health awareness (Δβ = 0.3717). The 95% confidence interval for this difference is [0.3074, 0.4360], which excludes 0, indicating that the difference is significant and supporting Hypotheses 2 and 3.

Additionally, following the recommendation of Cohen et al. [21], this study used the mean plus or minus one standard deviation for group participants. This approach was employed to illustrate the differences in the relationship between physical activity and exercise identity across varying levels of health awareness, as depicted in Fig 2.

## Discussion

### Theoretical contributions

Physical activity is a key antecedent variable in promoting individual subjective well-being, making it essential to investigate its underlying mechanisms. While previous studies have explored the mediating roles of psychological

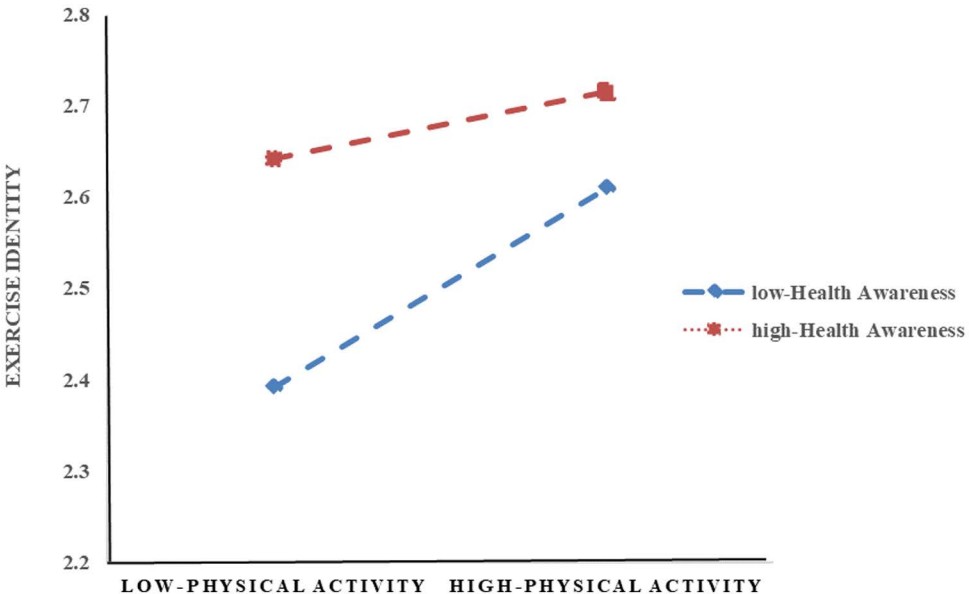

**Fig. 2. Moderating effect of health awareness on the relationship between physical activity and exercise identity.**

state, social capital, and interpersonal interactions, insufficient attention has been paid to the impact of individuals' identification with physical activity on subjective well-being. Similarly, health consciousness, which influences health behavior, has not been adequately examined in relation to the connection between physical activity and subjective well-being. This study fills these gaps by incorporating exercise identity as a mediating variable and health consciousness as a moderating variable and by constructing a moderated mediation model to explore how physical activity enhances subjective well-being. The findings offer significant theoretical contributions, enriching and expanding existing research.

First, this study demonstrates that exercise identity plays a crucial mediating role in the relationship between physical activity and subjective well-being. This finding suggests that an individual's identification with the value of physical activity is vital for enhancing well-being. By engaging in physical activity, individuals develop a sense of identity, affirming and shaping their self-concept. The process of finding, shaping, and affirming the "self" through physical activity and of gaining fulfillment, achievement, and belonging is key to improving subjective well-being. These findings expand the scope of research on the benefits of physical activity for well-being.

Second, this study finds that health consciousness positively moderates the relationship between physical activity and exercise identity as well as the mediating role of exercise identity in the relationship between physical activity and subjective well-being. Health consciousness indirectly influences individuals' physical activity behavior; the greater the level of health consciousness is, the stronger the motivation to engage in physical exercise. Individuals with greater health consciousness are more likely to recognize and experience the value of physical exercise, as they perceive its benefits for both physical and mental development. This enhanced awareness strengthens their exercise identity and, consequently, promotes their subjective well-being. These findings provide new insights into the mechanisms through which physical activity promotes well-being, filling gaps in prior research and offering novel strategies for future studies. Moreover, they further enrich the theoretical framework of physical activity and well-being.

In conclusion, this study provides deeper insights into the intrinsic relationship between physical activity and subjective well-being. It contributes new perspectives to the field and offers valuable evidence for future research.

## Practical significance

The effect of physical exercise on human well-being and its underlying mechanisms have become key areas of research. The value of physical activity in promoting physical and mental health, enhancing subjective well-being, and maintaining social security and stability is increasingly recognized. It is both important and necessary to explore in depth the mechanisms through which physical activity promotes well-being and to fully leverage the value and efficacy of sports. This study investigates the effects of physical activity on well-being and its underlying mechanisms, which holds significant practical value for advancing public fitness initiatives and enhancing public well-being.

First, this study reaffirms previous findings that physical exercise positively contributes to the enhancement of subjective well-being. Therefore, governments should continue to strengthen public sports services to meet both the material and psychological needs of the population. On the one hand, investments should be made to improve sports infrastructure and fitness facilities, alleviating the common issue of a shortage of accessible venues. On the other hand, scientific fitness services should be expanded, particularly through digital platforms. These services should promote the value of sports and health, fill the public's knowledge gaps regarding fitness, and raise health awareness. Doing so will not only enhance public enthusiasm for physical exercise but also foster a culture of health consciousness. Furthermore, local governments should guide individuals to engage in appropriate and scientifically informed physical activities that suit their personal characteristics. By experiencing the benefits of exercise, individuals can enhance their subjective well-being, fostering both personal growth and societal development.

Second, this study's exploration of the mediating role of exercise identity provides theoretical guidance for future public health and fitness management. Physical activity plays a crucial role in shaping exercise identity, suggesting that future research should further explore the general principles and mechanisms of exercise identity development. Promoting individuals' self-concept in the context of physical activity and strengthening both self-identity and sports value identity are important directions for future studies and practices.

Finally, this study reveals the positive moderating role of health consciousness in the relationship between physical activity and subjective well-being. Therefore, efforts should be made to further strengthen health education and promote greater health awareness. First, targeted health education programs should be developed for different population groups. Second, the curriculum and content of health education should be improved, and greater investments in training health education professionals should be made. Third, health education should be integrated into physical education curricula to raise students' health awareness, build a strong foundation for their future health, and improve overall health literacy. In short, health education efforts should follow a systematic plan and be implemented progressively to raise public health awareness and encourage proactive health behaviors. By fostering greater participation in physical exercise, the value and efficacy of sports can be fully realized, ultimately enhancing individuals' well-being and promoting healthy societal development.

## Research limitations and future directions

This study has several limitations. First, the data were collected primarily from adults in seven cities in China, which may introduce potential influences from the cultural or economic differences between regions. Additionally, the use of convenience sampling may result in selection bias, limiting the generalizability of the findings. Notably, 62.9% of the participants in the study sample held a bachelor's degree, and 15.6% held a master's degree, indicating a relatively high overall level of education. This aspect of the sample may make the findings more applicable to highly educated populations and less generalizable to groups with lower educational attainment. Future research could address this limitation by expanding the sample source to include individuals from more diverse educational backgrounds, thereby enhancing the external validity of the results. Second, while this study focused on the mediating role of exercise identity and the moderating role of health consciousness, other potential mediating and moderating variables may also influence the relationship between physical activity and subjective well-being. Since well-being is likely shaped by the interaction of multiple factors, future research should explore additional variables to provide a more comprehensive understanding. Finally, this study relied on

cross-sectional data, which may not fully capture the gradual process through which physical activity enhances subjective well-being. Future studies should employ longitudinal designs to better examine the long-term effects of physical activity on well-being over time.

## Conclusion

This study constructed a model examining the relationships between physical activity, health consciousness, exercise identity, and subjective well-being on the basis of data from 584 adult participants across seven Chinese cities. The findings indicate the following: 1. Physical activity has a significant positive effect on subjective well-being and on exercise identity. Exercise identity, in turn, significantly influences subjective well-being and serves as a key mediator in these relationships. 2. Health consciousness positively moderates the relationship between physical activity and exercise identity and enhances the mediating effect of exercise identity on the link between physical activity and subjective well-being.

## Supporting information

**S1 Data. The presentation of raw data.**
(XLSX)

## Author contributions

**Conceptualization:** Hong-cheng Cui.

**Data curation:** Yu Zhou.

**Formal analysis:** Hong-cheng Cui, Yu Zhou.

**Funding acquisition:** Hong-cheng Cui.

**Resources:** Hong-cheng Cui.

**Writing – original draft:** Yu Zhou.

**Writing – review & editing:** Yu Zhou.

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
