## [Decision Letter · Decision Letter 0]

27 Nov 2024

PONE-D-24-48825The Impact of Physical Activity on Subjective Well-Being: The Mediating Role of Exercise Identity and the Moderating Role of Health ConsciousnessPLOS ONE

Dear Dr. Zhou,

Thank you for submitting your manuscript to PLOS ONE. After careful consideration, we feel that it has merit but does not fully meet PLOS ONE’s publication criteria as it currently stands. Therefore, we invite you to submit a revised version of the manuscript that addresses the points raised during the review process.

We look forward to receiving your revised manuscript.

Kind regards,

Henri Tilga, PhD

Academic Editor

PLOS ONE

Journal Requirements:

4. We are unable to open your Supporting Information file “data.sav.” Please kindly revise as necessary and re-upload.

Reviewers' comments:

Reviewer's Responses to Questions

**Comments to the Author**

1. Is the manuscript technically sound, and do the data support the conclusions?

Reviewer #1: Partly

2. Has the statistical analysis been performed appropriately and rigorously? 

Reviewer #1: Yes

3. Have the authors made all data underlying the findings in their manuscript fully available?

Reviewer #1: Yes

4. Is the manuscript presented in an intelligible fashion and written in standard English?

Reviewer #1: No

5. Review Comments to the Author

Reviewer #1: Research topic selection is meaningful, but there are some problems in the paper that need to be improved:

1. The basic characteristics of the subjects are described too little, such as occupation, family or personal income, and marital status. This information needs to be supplemented.

2. he hypothesis of Figure 1 is flawed. Why is there a modulatory effect only assumed for the mediating path, but not for the direct path? A schematic representation of the moderating effect of only one pathway is drawn, and it should be assumed that health awareness has a moderating effect on all three pathways.

3. Health identity has a complete mediating effect, and health awareness should moderate the two mediating pathways.

4. What is the difference between MODEL 1 and MODEL 4 in Table 1? Model 1 has 4 factors, right? Model 4 only has one factor? It needs to be explained further.

5. The three variables of SEX, AGE and EDUCATION in Table 2 should be used as adjusted confounding variables, rather than as the main analysis variables. It is better to do partial correlation analysis with adjusted confounding variables.

6. In Table 2, the correlation between physical activity, health awareness and health identity is close to 0.6, and the variables are almost interchangeable. Is there a problem in the selection of the scale or the measurement process?

7. Neither Figure 1 nor Figure 2 is clearly expressed.

8.62.9% are bachelor degree and 15.6% are master degree. The study population was highly educated and the extrapolation of the findings was severely limited and not mentioned in the Limitation.

6. PLOS authors have the option to publish the peer review history of their article (what does this mean? ). If published, this will include your full peer review and any attached files.

**Do you want your identity to be public for this peer review?** For information about this choice, including consent withdrawal, please see our Privacy Policy .

Reviewer #1: No

---

## [Author Response · Author response to Decision Letter 1]

16 Dec 2024

Response to Reviewers

We sincerely thank the editor and the reviewer for their comments and advice, which we have used to improve the manuscript. Point-by-point responses to each comment follow below. The line numbers refer to the revised manuscript.

Journal Requirements:

Comment 1: Please ensure that your manuscript meets PLOS ONE's style requirements, including those for file naming. The PLOS ONE style templates can be found at https://journals.plos.org/plosone/s/file?id=wjVg/PLOSOne_formatting_sample_main_body.pdf and https://journals.plos.org/plosone/s/file?id=ba62/PLOSOne_formatting_sample_title_authors_affiliations.pdf.

Response:

Thank you for your comment. We have carefully reviewed the PLOS ONE style requirements and revised our manuscript to ensure full compliance with the guidelines. The file naming conventions have also been updated accordingly. Additionally, we have consulted the style templates provided at the links and formatted the manuscript to align with these templates.

Comment 2: Please include your full ethics statement in the ‘Methods’ section of your manuscript file. In your statement, please include the full name of the IRB or ethics committee who approved or waived your study, as well as whether or not you obtained informed written or verbal consent. If consent was waived for your study, please include this information in your statement as well.

Response:

Thank you for your valuable feedback. We have updated the ‘Methods’ section of our manuscript to include the full ethics statement. The statement now specifies the name of the Institutional Review Board (IRB) that approved the study—Guangzhou Sports University Ethics Committee. It also clarifies that informed written consent was obtained from all participants (p. 6, 126–130).

Comment 3: Please include captions for your Supporting Information files at the end of your manuscript, and update any in-text citations to match accordingly. Please see our Supporting Information guidelines for more information: http://journals.plos.org/plosone/s/supporting-information.

Response:

Thank you for your comment. We have revised the manuscript according to the reviewers' comments as requested.

Comment 4: We are unable to open your Supporting Information file “data.sav.” Please kindly revise as necessary and re-upload.

Response:

Thank you for bringing this issue to our attention. Upon review, we have converted the file to Excel format.

Response to Reviewer

Comment 1: Is the manuscript presented in an intelligible fashion and written in standard English? PLOS ONE does not copyedit accepted manuscripts, so the language in submitted articles must be clear, correct, and unambiguous. Any typographical or grammatical errors should be corrected at revision, so please note any specific errors here. Reviewer #1: No

Response:

Thank you for your feedback. The manuscript was edited for proper English language, grammar, punctuation, spelling, and overall style by one or more of the highly qualified English speaking editors at SNAS. All typographical and grammatical errors identified have been corrected. Additionally, we have conducted a thorough review of the language to ensure that it adheres to the standards of academic English.

Comment 2: The basic characteristics of the subjects are described too little, such as occupation, family or personal income, and marital status. This information needs to be supplemented.

Response:

Thank you for your valuable comment. In the revised manuscript, we have supplemented the description of the subjects’ basic characteristics by including information on their occupation, family or personal income, and marital status. This additional information is now detailed in the ‘Participants’ subsection of the Methods section (p. 6, 117–125).

Comment 3: The hypothesis of Figure 1 is flawed. Why is there a modulatory effect only assumed for the mediating path, but not for the direct path? A schematic representation of the moderating effect of only one pathway is drawn, and it should be assumed that health awareness has a moderating effect on all three pathways.

Response:

Thank you for your insightful comment regarding the hypotheses in Figure 1. We agree that it is important to consider the potential moderating effects of health awareness comprehensively. In the revised manuscript, we have supplemented the theoretical rationale to justify why the moderating effect of health awareness was initially hypothesized only for the mediating path (p. 5, 85-101).

Comment 4: Health identity has a complete mediating effect, and health awareness should moderate the two mediating pathways.

Response: Thank you for your suggestion. We have revised the manuscript accordingly (p. 5, 85-101).

Comment 5: What is the difference between MODEL 1 and MODEL 4 in Table 1? Model 1 has 4 factors, right? Model 4 only has one factor? It needs to be explained further.

Response:

Thank you for pointing out the need for further clarification regarding the differences between MODEL 1 and MODEL 4 in Table 1. We have addressed this issue in the revised manuscript. Specifically, we added a detailed explanation of the differences between these models, including their respective factors and theoretical implications (p. 9, 180-184).

Comment 6: The three variables of SEX, AGE, and EDUCATION in Table 2 should be used as adjusted confounding variables, rather than as the main analysis variables. It is better to do partial correlation analysis with adjusted confounding variables.

Response:

Thank you for this valuable comment. We have carefully revised the manuscript to address your concerns. Specifically, we have reanalyzed the data to treat SEX, AGE, and EDUCATION as adjusted confounding variables rather than as the main analysis variables. Partial correlation analyses have been conducted accordingly, and the results have been updated in Table 2 to reflect this adjustment (p. 9).

Comment 7: In Table 2, the correlation between physical activity, health awareness, and health identity is close to 0.6, and the variables are almost interchangeable. Is there a problem in the selection of the scale or the measurement process?

Response:

Thank you for raising this important concern. To address it, we conducted a thorough review of the scales used and the measurement process to ensure their appropriateness. We confirmed that the scales for physical activity, health awareness, and health identity were selected on the basis of established theoretical frameworks and validated in prior studies.

Additionally, we have included a detailed explanation in the manuscript to clarify the rationale for scale selection, as well as the steps taken to minimize measurement overlap. (pp. 9-10, 190-210)

Comment 8: Neither Figure 1 nor Figure 2 is clearly expressed.

Response:

Thank you for your valuable feedback. We have carefully reviewed Figures 1 and 2 to address the clarity issues you raised.

1. Figure 1: We have provided additional explanations for Table 1 in the manuscript to clarify the relationships and variables depicted in Figure 1. These modifications aim to ensure that the figure aligns more closely with the theoretical framework and is easier to interpret.

2. Figure 2: We have redrawn Figure 2 to improve its clarity and visual representation. The updated figure uses a more structured layout, clearer labeling, and consistent formatting to increase readability.

Comment 9: 62.9% are bachelor degree and 15.6% are master degree. The study population was highly educated, and the extrapolation of the findings was severely limited and not mentioned in the Limitation.

Response:

Thank you for pointing out this important limitation of our study. We have addressed your concern by adding a detailed explanation of the limitation regarding the high educational level of the study population. This discussion highlights how the findings might not be fully generalizable to populations with different educational backgrounds (p. 18, 345-352).

---

## [Decision Letter · Decision Letter 1]

6 Jan 2025

PONE-D-24-48825R1The Impact of Physical Activity on Subjective Well-Being: The Mediating Role of Exercise Identity and the Moderating Role of Health ConsciousnessPLOS ONE

Dear Dr. Zhou,

Thank you for submitting your manuscript to PLOS ONE. After careful consideration, we feel that it has merit but does not fully meet PLOS ONE’s publication criteria as it currently stands. Therefore, we invite you to submit a revised version of the manuscript that addresses the points raised during the review process.

We look forward to receiving your revised manuscript.

Kind regards,

Henri Tilga, PhD

Academic Editor

PLOS ONE

Journal Requirements:

Reviewers' comments:

Reviewer's Responses to Questions

**Comments to the Author**

1. If the authors have adequately addressed your comments raised in a previous round of review and you feel that this manuscript is now acceptable for publication, you may indicate that here to bypass the “Comments to the Author” section, enter your conflict of interest statement in the “Confidential to Editor” section, and submit your "Accept" recommendation.

Reviewer #2: All comments have been addressed

2. Is the manuscript technically sound, and do the data support the conclusions?

Reviewer #2: Yes

3. Has the statistical analysis been performed appropriately and rigorously? 

Reviewer #2: Yes

4. Have the authors made all data underlying the findings in their manuscript fully available?

Reviewer #2: Yes

5. Is the manuscript presented in an intelligible fashion and written in standard English?

Reviewer #2: Yes

6. Review Comments to the Author

Reviewer #2: 1. Method: In the measures section, the name of the scale used can be listed as a single line of subheadings. Common methodological biases can be placed in the results section. The content of reliability and validity analysis can be separately placed in the introduction of relevant scales in the measures.

2.Conclusion and discussion: The writing format of this part does not conform to the standard of writing academic papers. Usually write the discussion first and the conclusion last. In addition, the content of the conclusion mainly describes the content of the research results, rather than the research conclusion.

7. PLOS authors have the option to publish the peer review history of their article (what does this mean? ). If published, this will include your full peer review and any attached files.

**Do you want your identity to be public for this peer review?** For information about this choice, including consent withdrawal, please see our Privacy Policy .

Reviewer #2: No

---

## [Author Response · Author response to Decision Letter 2]

18 Feb 2025

Response to Reviewers

We sincerely thank the editor and the reviewer for their comments and advice, which we have used to improve the manuscript. Point-by-point responses to each comment follow below. The line numbers refer to the revised manuscript.

Journal Requirements:

Comment 1: Please review your reference list to ensure that it is complete and correct. If you have cited papers that have been retracted, please include the rationale for doing so in the manuscript text, or remove these references and replace them with relevant current references. Any changes to the reference list should be mentioned in the rebuttal letter that accompanies your revised manuscript. If you need to cite a retracted article, indicate the article’s retracted status in the References list and also include a citation and full reference for the retraction notice.

Response:

We have carefully reviewed all references to ensure their completeness and accuracy. No retracted papers were cited in our manuscript. Therefore, no changes were necessary. If further adjustments are required, we are happy to make them.

Response to Reviewer 2

Comment 1: Method: In the measures section, the name of the scale used can be listed as a single line of subheadings. Common methodological biases can be placed in the results section. The content of reliability and validity analysis can be separately placed in the introduction of relevant scales in the measures.

Response:

We sincerely appreciate the reviewer’s valuable suggestions regarding the structure of the Measures section. In response, we have made the following revisions:

1. We have listed the names of the scales as single-line subheadings to improve clarity and readability (p. 7, 114, 144, 148, 152).

2. Common methodological biases have been moved to the Results section, as suggested (p. 9, 181–186).

3. The reliability and validity analyses remain in the Methods section but have been placed after the introduction of the scales to ensure a logical flow and better distinction between the measurement tools and their psychometric properties (p. 8, 158).

These changes enhance the overall structure and clarity of the manuscript while maintaining coherence in presenting the methodological framework. We appreciate the reviewer’s constructive feedback and the opportunity to improve our work.

Comment 2: Conclusion and discussion: The writing format of this part does not conform to the standard of writing academic papers. Usually write the discussion first and the conclusion last. In addition, the content of the conclusion mainly describes the content of the research results, rather than the research conclusion.

Response:

Thank you for your valuable feedback regarding the structure of the Discussion and Conclusion sections. In response to your comment, I have made the following adjustments:

1. I have restructured the manuscript to place the Discussion section before the Conclusion, in accordance with academic writing standards (p. 14, 249, 250, 287, 331).

2. I have revised the Conclusion to focus on summarizing the research findings rather than reiterating the content of the study. The revised conclusion now presents the key results and insights derived from the study, offering a clear and concise summary (p. 19, 353–362).

---

## [Decision Letter · Decision Letter 2]

11 Mar 2025

The Impact of Physical Activity on Subjective Well-Being: The Mediating Role of Exercise Identity and the Moderating Role of Health Consciousness

PONE-D-24-48825R2

Dear Dr. Zhou,

We’re pleased to inform you that your manuscript has been judged scientifically suitable for publication and will be formally accepted for publication once it meets all outstanding technical requirements.

Kind regards,

Henri Tilga, PhD

Academic Editor

PLOS ONE

Additional Editor Comments (optional):

Reviewers' comments:

Reviewer's Responses to Questions

**Comments to the Author**

1. If the authors have adequately addressed your comments raised in a previous round of review and you feel that this manuscript is now acceptable for publication, you may indicate that here to bypass the “Comments to the Author” section, enter your conflict of interest statement in the “Confidential to Editor” section, and submit your "Accept" recommendation.

Reviewer #2: All comments have been addressed

2. Is the manuscript technically sound, and do the data support the conclusions?

Reviewer #2: (No Response)

3. Has the statistical analysis been performed appropriately and rigorously? 

Reviewer #2: (No Response)

4. Have the authors made all data underlying the findings in their manuscript fully available?

Reviewer #2: (No Response)

5. Is the manuscript presented in an intelligible fashion and written in standard English?

Reviewer #2: (No Response)

6. Review Comments to the Author

Reviewer #2: (No Response)

7. PLOS authors have the option to publish the peer review history of their article (what does this mean? ). If published, this will include your full peer review and any attached files.

**Do you want your identity to be public for this peer review?** For information about this choice, including consent withdrawal, please see our Privacy Policy .

Reviewer #2: No

---

## [Editor Report · Acceptance letter]

PONE-D-24-48825R2

PLOS ONE

Dear Dr. Zhou,

I'm pleased to inform you that your manuscript has been deemed suitable for publication in PLOS ONE. Congratulations! Your manuscript is now being handed over to our production team.

Kind regards,

on behalf of

Dr. Henri Tilga

Academic Editor

PLOS ONE